

**SEAMUS (v1.0): a Δ[14]C-enabled, single-specimen sediment accumulation simulator**
Bryan C. Lougheed[1,2]
1. Department of Earth Sciences, Uppsala University, Uppsala, Sweden.
2. Laboratoire d'Océanologie et de Géosciences, Université du Littoral Côte d'Opale, Wimereux,
France.
Corresponding author: B.C. Lougheed (bryan.lougheed@geo.uu.se)
**Abstract**
The systematic bioturbation of single particles (such as foraminifera) within deep-sea sediment
archives leads to the apparent smoothing of any temporal signal as record by the downcore,
discrete-depth mean signal. This smoothing is the result of the systematic mixing of particles from a
wide range of depositional ages into the same discrete depth interval. Previous sediment models
that simulate bioturbation have specifically produced an output in the form of a downcore, discrete-
depth mean signal. Palaeoceanographers analysing the distribution of single foraminifera specimens
from sediment core intervals would be assisted by a model that specifically evaluates the effect of
bioturbation upon single specimen populations. Taking advantage of recent increases in computer
memory, the single-specimen SEdiment AccuMUlation Simulator (SEAMUS) was created in Matlab,
whereby large arrays of single specimens are simulated. This simulation allows researchers to
analyse the post-bioturbation age heterogeneity of single specimens contained within discrete-
depth sediment core intervals, and how this heterogeneity is influenced by changes in sediment
accumulation rate (SAR), bioturbation depth (BD) and species abundance. The simulation also
assigns a realistic [14]C activity to each specimen, by considering the dynamic Δ[14]C history of the Earth
and temporal changes in reservoir age. This approach allows for the quantification of possible
significant artefacts arising when [14]C dating multi-specimen samples with heterogeneous [14]C activity.
Users may also assign additional desired carrier signals to specimens (e.g., stable isotopes, trace
elements, temperature, etc.) and consider a second species with an independent abundance. Finally,
the model can simulate a virtual palaeoceanographer by randomly picking whole specimens
(whereby the user can set the percentage of older, 'broken' specimens) of a prescribed sample size
from discrete depths, after which virtual laboratory [14]C dating and [14]C calibration is carried out
within the model.



**1.0 Introduction**

Deep-sea sediment archives provide valuable insight into past changes in ocean circulation and global climate. The most often studied carrier vessels of the climate signal are the calcite tests of foraminifera. The tests of these organisms incorporate isotopes and trace elements of the ambient water at the time of calcification, before sinking to the seafloor sediment archive after death. Each discrete-depth interval of a sediment core (typically 1 cm core slices) retrieved from the sea floor can contain many thousands of specimens. Researchers have typically had to combine many tens or hundreds of single tests into a single sample for successful analysis using mass spectrometry. Furthermore, post-depositional sediment mixing (e.g. bioturbation (Berger and Heath, 1968)) of deep-sea sediment means that foraminifera specimens of vastly differing ages can be mixed into the same discrete-depth interval. The main consequence of this mixing is that a downcore, discrete-depth multi-specimen reconstruction of a specific climate proxy will appear to be strongly smoothed out (on the order of multiple centuries or millennia) when compared to the original temporal signal (Pisias, 1983; Schiffelbein, 1984; Bard et al., 1987). Moreover, machine analysis of multi-specimen samples will only report the mean value and machine error, thus hiding the true distribution of values within the sample. Advances in mass spectrometry eventually allowed the analysis of single specimens (Killingley et al., 1981) and, since single specimens capture a single year/season of the climate signal, researchers can study the full distribution of isotope or trace element values obtained from single specimens contained within various discrete depths of sediment cores to make inferences regarding variability in climate, habitat or specimen morphology for various specific time periods during the Earth's history (Spero and Williams, 1990; Tang and Stott, 1993; Billups and Spero, 1996; Ganssen et al., 2011; Wit et al., 2013; Ford et al., 2015; Metcalfe et al., 2015, 2019b; Ford and Ravelo, 2019). However, the accuracy with which the aforementioned studies can quantify time-specific variation for a particular climate period, habitat or morphological variable is strongly dependent upon the constraint of the age range of the specimens contained within a given discrete-depth interval. The aforementioned studies still rely strongly upon the mean depth age method to assign an age range to all specimens contained within a discrete depth interval, and previous models of single specimen analysis in sediment cores do not include bioturbation (Thirumalai et al., 2013; Fraass and Lowery, 2017). Such an approach can be problematic if, to give but one example, an assumed Holocene age 1-cm slice of sediment core were to also contain a significant number of Late Glacial specimens, which could lead to a spurious interpretation of Holocene climate variability. Ultimately, this problem can be circumvented through the application of paired analysis of both radiocarbon ($^{14}$C) and stable isotopes on single specimens (Lougheed et al., 2018), but the current mass requirements of $^{14}$C accelerated mass spectrometry (AMS) means that such a method is



currently limited to very large specimens (>100 μg), whereas most planktonic foraminifera used in
palaeoceanography are of an order of magnitude smaller. Until such time that single specimen $^{14}$C
methods become systematically applicable to planktonic specimens, and for periods older than the
analytical limit of $^{14}$C dating (>50 ka), a sediment accumulation model specifically designed for the
analysis of single specimens can help shed light on the age distributions planktonic foraminifera
contained within discrete depths.
Using a model to quantify the distribution of specimen ages within discrete-depth sediment intervals
is also important for $^{14}$C dating applied to multi-specimen samples, which can be expected to have
heterogeneous radiocarbon ($^{14}$C) activity. This heterogeneity is governed by the Earth's dynamic Δ$^{14}$C
history, temporal changes in species abundance, sediment accumulation rate (SAR) and in local $^{14}$C
reservoir age. Temporal changes in $^{14}$C heterogeneity have the potential to induce downcore age-
depth artefacts when $^{14}$C analysis and $^{14}$C calibration are applied to multi-specimen samples. The
ability to make a quantitative estimate of downcore changes in the $^{14}$C heterogeneity and its effect
upon $^{14}$C dating would help to improve Late Glacial and Holocene geochronologies for deep-sea
sediment archives.
Here, the Δ$^{14}$C-enabled single-specimen SEdiment AccuMUlation Simulator (SEAMUS) is presented.
This model takes advantage of advances in computing power to simulate a large array of single
specimens. Such an approach allows for a relatively straightforward execution of transient runs with
temporally dynamic time series inputs for sediment accumulation rate (SAR), species abundance,
bioturbation depth (BD), $^{14}$C reservoir age, Δ$^{14}$C and any desired carrier signal(s). Single specimen
populations are essentially transferred from the time domain to the depth domain, thus simulating
the sedimentation history of the resulting sediment archive. The distribution of discrete depth single
specimen true age, $^{14}$C activity, bioturbation history (number of bioturbation cycles), and carrier
signal can subsequently be investigated and relationships with the dynamic input parameters can be
explored. Subsequently, users can subject the simulated sediment archive to a picking procedure
(with a prescribed number of randomly picked whole specimens per sample) to create virtual
subsamples from each discrete core depth, whereby one can also consider the presence of broken
(non-picked) specimens, which have been through more bioturbation cycles and are therefore older.
From these virtual subsamples, mean carrier signal values and species abundances can be calculated,
allowing users to evaluate their downcore core reconstructions for the possible presence of
artefacts. Furthermore, these virtual subsamples can be used to calculate virtual laboratory $^{14}$C
dates, which are subsequently calibrated using the *MatCal* (Lougheed and Obrochta, 2016)
calibration software. Calibrated age distributions for a discrete depth can be compared to their



associated simulated true age distribution, thus evaluating the accuracy of the $^{14}$C dating and
calibration process.
**2.0 Model description**
**2.1 Bioturbation understanding and previous models**
The most commonly used mathematical model of bioturbation in deep-sea sediments is the so-
called Berger-Heath bioturbation model, which assumes a uniform an instantaneous (on geological
timescales) mixing of the bioturbation depth (BD), the uppermost portion of a sediment archive
where oxygen availability allows for the active bioturbation of sediments (Berger and Heath, 1968;
Berger and Johnson, 1978; Berger and Killingley, 1982). Observations of uniform mean age in the
uppermost intervals of sediment archives do indeed support this mixing model (Peng et al., 1979;
Boudreau, 1998), and the BD itself has been shown to be related to the organic carbon flux at the
seafloor (Trauth et al., 1997). Researchers wishing to carry out transient bioturbation simulations
with dynamic input parameters have incorporated the Berger-Heath mathematical model into their
computer models to, most notably the FORTRAN77 model TURBO (Trauth, 1998), its updated
MATLAB version TURBO2 (Trauth, 2013) and the more recent R model Sedproxy (Dolman and
Laepple, 2018). In the case of TURBO2, the user inputs a number of idealised, non-bioturbated
stratigraphical levels with assigned age, depth, carrier signal and abundance. Subsequently, TURBO2
outputs the bioturbated carrier signal and abundance values corresponding to the inputted
stratigraphic levels. Consequently, TURBO2 is of most interest for researchers who would like to
understand the perturbation of the mean downcore signal. Sedproxy allows the user to input a
climate data in the time domain, along with sediment core variables (such as SAR and BD), after
which mathematical computations are used to produce the equivalent bioturbated climate data also
in the time domain, whereby single specimen distributions can also be quasi-inferred.
**2.2 The SEAMUS model**
**2.2.1 Short description of the model**
The SEAMUS simulation is an iterative model that actively simulates the sedimentation process of
single specimens on a per timestep basis, whereby input data in the time domain is converted into
the core depth domain. For each timestep, a number of new specimens are added to the top of the
simulated core, with bioturbation subsequently being carried out. SEAMUS uses the sediment core
and species abundance variables inputted in the time domain (SAR in the form of an age-depth
model, BD vs time, species abundance vs time) to simulate a number of new single specimens per
timestep. Each of these specimens are assigned an age, $^{14}$C activity, reservoir age and carrier signal





corresponding to the timestep. Subsequently, the new specimens are added to the top of the
existing core, after which bioturbation is carried out. The simulation takes advantage of recent
increases in computer memory capacity to keep track of the depths, ages, [14]C activities, species
types and number of bioturbation cycles for all single specimens in the simulation. Such an
approach, which is optimised for single specimens, allows the user to use logical indexing to quickly
access all variables for given single specimens for given depths, ages and/or species.
The SEAMUS simulation is broken down into two main functions that the user can call. The first
function *seamus_run*, carries out the actual single specimen sedimentation simulation based on the
input parameters designated by the user. The second function, *seamus_pick*, can be best described
as a 'virtual palaeoceanographer', in that it carries out downcore analysis of the simulated sediment
core, including discrete-depth sample picking, calculation of sub-sample mean carrier signals, [14]C
analysis by virtual AMS, [14]C calibration, etc. The *seamus_run* and *seamus_pick* functions, as well as
their associated input and output variables, are detailed in sections 2.3.2 and 2.3.3.
**2.2.2 The sediment core simulation (*seamus_run*)**
The *seamus_run* module uses the required and optional input parameters specified by the user
(Table S1) to synthesise $n$ number of single specimens being net-added to the historical layer of the
sediment core per simulation timestep, whereby $n$ is scaled to the capacity of the synthetic sediment
archive being simulated (input variable *fpcm*) and to the SAR of the timestep as predicted by an
inputted age-depth relationship. The simulation creates large single specimen arrays of matching
dimensions for age (corresponding to the timestep), 'unbioturbated' sediment depth (according to
the age-depth input), as well as a [14]C age (in [14]C yrs) and [14]C activity (in $f$MC). The user also has the
option to input a [14]C blank value. Furthermore, all single specimens can be assigned carrier signal
values. It should be noted that the user is not required to enter input values for every timestep: for
example, an age-depth relationship can simply be inputted with a handful of data points and the
model will automatically linearly interpolate to create age and depth values for every simulation
timestep. The same principle holds true for other temporally dynamic inputs such as species
abundance, reservoir age and carrier signals.
After the creation of all new single specimens within the synthetic core, a per timestep bioturbation
simulation of the depth array is carried out. Specifically, for each timestep the depth values
corresponding to all simulated specimens within the timestep-specific active BD are each assigned a
new depth by way of uniform random sampling of the BD interval. In this way, uniform mixing of
specimens within the BD is simulated following established understanding of bioturbation. The per



timestep bioturbation simulation is carried out in *seamus_run* as follows; first, the simulation finds
the indices for all specimen depth values present in the contemporaneous BD:
```
ind = find(depths >= addepths(s) & depths < addepths(s) + biodepths(s))
```

Where *addepths(s)* is the depth corresponding to the age for timestep *s*, i.e. *addephts(s)* is
analogous to the timestep's core top; and where *biodepths(s)* is the BD corresponding to the age for
timestep *s.*
Subsequently, all specimen depth values corresponding to the active BD are assigned new depth
values by uniform random sampling of the active BD itself:
```
depths(ind) = rand(length(ind),1)*biodepths(s) + addepths(s)
```

The simulation uses a simple counter array to keep track of how many times each single specimen
has been subjected to a bioturbation cycle:
```
cycles(ind) = cycles(ind) + 1
```

All of the aforementioned processes are repeated for every simulation timestep until such point that
the end of the age-depth input (i.e. the final core top) is reached. Currently, the simulation carries
out bioturbation according to a per timestep uniform random sampling, but users wishing to
experiment with other types of bioturbation (i.e. partial bioturbation, etc.) can modify the
aforementioned lines of the script.
It is recommended that users initiate the *seamus_run* simulation with sufficient spinup time. The
necessary spin-up time can vary dependent upon the SAR and BD being studied, but for most
applications (SAR >5 cm/ka), a spin-up time of at least 20 ka should suffice. In other words, if one is
studying a period of interest that commences at 50 ka ago, then the simulation can be started at 70
ka ago. The required input parameters should be inputted in the command line as follows:
```
seamus_run(simstart, siminc, simend, btinc, fpcm, realD14C, blankbg,
```
```
adpoints, bdpoints, savename)
```

Optional parameters can be additionally specified as follows, e.g. in the case of including the matrix
*matrixname* containing temporal changes in reservoir age for Species A:
```
seamus_run(simstart, siminc, simend, btinc, fpcm, realD14C, blankbg,
```
```
adpoints, bdpoints, savename, 'resageA', matrixname)
```

The *seamus_run* module outputs a .mat file containing a number of very large 1 arrays of the same
dimension, whereby each position in each array corresponds to the same simulated single





specimens. Output variables are detailed in Table S2. To improve performance and ease of use, all
output variables are simulated for all single specimens. For example, carrier signals specific to
Species A (carrierA) are simulated for both Species A and Species B. As all output variables are of the
same dimension, one can easily isolate the carrierA signals specific to the specimens of Species A
(*types* value of 0) using logical indexing:
`carrierA(types == 0 , :)`
and from a specific core depth interval (e.g. between 16 and 17 cm):
`carrierA(types == 0 & depths >= 16 & depths < 17 , :)`
**2.2.3 Virtual picking of the simulated sediment core (*seamus_pick*)**
The *seamus_pick* module carries out a simple picking simulation upon the simulated core generated
by *seamus_run*. Users are able to set a specific sample size (i.e. the number of single specimens to
be randomly picked per sample), sample picking interval (i.e. core slice thickness) and optionally
include information about the amount of broken/non-whole specimens. The latter parameter is set
as a fraction of the entire specimen population, whereby the fraction of the population that has
been through the most bioturbation cycles is assumed to be broken. For example, if the user sets the
fraction of broken specimens to 0.25, then the simulation will only randomly pick from the specimen
population with bioturbation cycles between the 1st and 75th percentiles. In this way, the preference
of a palaeoceanographer to pick whole specimens is simulated.
Within *seamus_pick*, virtual $^{14}$C laboratory analysis is carried out on the picked subsamples by
calculating the mean $^{14}$C activity (in fMC), after which the resulting mean fMC value is converted into
$^{14}$C age (in $^{14}$C yr). A realistic measurement error is also assigned to to each $^{14}$C age, whereby a late
Holocene $^{14}$C age is assumed to have a measurement error of ±30 $^{14}$C yr, and a $^{14}$C age of just above
the blank value is assumed to have an error of ±200 $^{14}$C yr. Measurement errors for ages in between
are linearly scaled to $^{14}$C activity. Using the *MatCal* (Lougheed and Obrochta, 2016) calibration
software, $^{14}$C ages and errors are calibrated inline, after the application of a user-prescribed
calibration curve and downcore reservoir age.
The *seamus_pick* function is called from the command line:
`seamus_pick(matfilein, matfileout, calcurve, pickint, Apickfordate,`
`Bpickfordate)`
Optional parameters can be additionally specified as follows, e.g. in the case of including the matrix
*matrixname* containing downcore changes in the fraction of broken specimens in Species A:



```
seamus_pick(matfilein, matfileout, calcurve, pickint, Apickfordate,
Bpickfordate, 'Abroken', matrixname)
```

**2.2.4 Suggested input data**
Users are free to use any input data they please, so long as it abides to the specified requirements as
listed in the function documentation, as well as in Tables S1 and S3. This freedom can allow users to
carry out abstract modelling experiments to increase understanding of the relationship between
input variables, the resulting downcore single specimen vales and trends in downcore discrete-depth
means. Alternatively, users can try to forward model an actual sediment core record in order to
investigate for the possible presence of bioturbation/abundance artefacts within their sediment core
record. An existing age-depth model of a sediment core could be used as the dynamic age-depth
input for the SEAMUS simulation, although users must be aware that age-depth models may
themselves contain artefacts caused by the interaction between bioturbation and abundance. Data
regarding downcore abundance estimates could be used as abundance estimates, but similarly,
users should be aware that observed downcore abundance in the core depth domain is not the same
as original abundance in the time domain. Users could, therefore, experiment in using multiple
temporal abundance and bioturbation depth combinations as simulation input, and rerunning the
simulation with different temporal abundance and bioturbation depth combinations until such time
that generated abundance data in depth is similar to the observed abundance in depth. Input
climate data for simulations could be based on multiple experimental, fictional scenarios, geological
records, or generated from isotope-enabled climate models (Roche, 2013) coupled to, for example, a
foraminifera ecology model such as FORAMCLIM (Lombard et al., 2011) or FAME (Roche et al., 2018;
Metcalfe et al., 2019a), to produce a fully parameterised "climate to sediment core" model
workflow.
**3.0 Model Evaluation**
**3.1 Comparison with TURBO2**
In order to evaluate the performance of the SEAMUS model, it is compared here to the output of the
established TURBO2 bioturbation model (Trauth, 2013), which was also authored in Matlab. The
most notable difference between SEAMUS and TURBO2 is that the latter outputs data in the form of
the perturbation of the mean downcore signal, whereas SEAMUS takes advantage of recent
increases in available computer memory to store and output a very large array of single elements
(foraminifera specimens). The two models can be compared, therefore, by comparing the mean
downcore output from TURBO2 with the SEAMUS downcore mean value derived from discrete-



depth single specimen populations. To achieve this comparison, the NGRIP Greenland ice core $\delta^{18}$O
record on the GICC05 timescale (North Greenland Ice Core Project members, 2004; Rasmussen et al.,
2014; Seierstad et al., 2014) is used as a reference signal to represent the 'unbioturbated' climate
signal (Fig. 1a). This 50 year temporal resolution signal is subsequently inputted into both SEAMUS
and TURBO2 using identical run conditions comprising of a constant SAR of 10 cm/ka, a constant BD
of 10 cm and a single foraminiferal species with a constant abundance. The SEAMUS simulation is
run using a 10 year timestep. The TURBO2 and SEAMUS core simulations (i.e. single specimens in the
case of SEAMUS) are directly assigned the oxygen isotope values from the NGRIP record. One would
obviously not expect that foraminifera in the open ocean would have the same oxygen isotope
values as an ice sheet record (due to fractionation effects, habitat effects, oceanographic effects,
seasonal overprint, etc), but the purpose here is simply to compare the output of the respective
bioturbation algorithms in SEAMUS and TURBO2 using some kind of high-temporal resolution
climatic input signal. Furthermore, using the NGRIP record allows for the isolation of the
bioturbation effect upon a hypothesised single specimen record. The respective mean downcore
bioturbated signals produced by SEAMUS and TURBO2 are shown in Fig. 1b and exhibit a significant
correlation ($r^2$ = 0.99, p < 0.01), indicating that the SEAMUS approach is incorporating the same
understanding of bioturbation as TURBO2.
**3.2 Processing speed and computing requirements**
Where possible, the processing of variables for simulation timesteps has been vectorised (i.e. not
processed within an iterative loop), in order to maximise processing speed. For example, the per
timestep assignment of single specimen arrays corresponding to ages and carrier signals all occurs
within fully vectorised code. However, the bioturbation simulation (i.e. the bioturbation of the
assigned depth values) is not vectorised and is carried out within a single-thread iterative loop, due
to each iteration of the bioturbation simulation being dependent upon the results of the previous
iteration. In order to optimise the processing time on 64-bit computers, all arrays are stored as 64-
bit. Should the user wish to save memory, it is possible to select the *do32bit* option when accessing
*seamus_run* from the command line (see Table S1). Indicative run times and memory use are shown
in Table 1.
The SEAMUS model was developed in Matlab 2017b. The *seamus_run* module can be run using the
basic Matlab environment, with no extra toolboxes. The *seamus_pick* module runs more efficiently
when the Statistics and Machine Learning toolbox (specifically, the prctile function) is installed, but
when it is detected that users do not have access to that toolbox, *seamus_pick* will revert to using a
modified version of the equivalent function in Octave (Kienzle, 2001), which has been embedded




into the script. The *seamus_pick* function also requires the Matcal (Lougheed and Obrochta, 2016)
[14]C calibration script, which has been included in the SEAMUS download package.

**4.0 Potential model applications**

**4.1 Analysing downcore specimen population distributions**

As outlined in the introduction, advances in mass spectrometry have allowed for routine single
specimen analysis, which has led to increased interest in using analysis of single specimen
populations from discrete depths as a potentially powerful tool with which to reconstruct past
changes in climate variability. This application of this tool, however, still relies upon median
downcore age by assigning an age estimate to all single specimens from a single depth. Climate
variability/seasonality interpretations are clouded, therefore, when single specimens from a wide
range of ages are mixed into the same depth, especially if the interpretation relies upon detecting
extreme climate events in the form of single specimen outliers. Using the previously described
(Section 3.1; Fig 1b) SEAMUS simulation, it is possible to construct a probability heatmap and 95.45%
intervals for the single specimen $\delta^{18}O$ (Fig. 2a) data. The shape and range of these 95.45% intervals
relative to a glacial-interglacial change is similar to what has been previously calculated by
(Schiffelbein, 1986), albeit in the case of the Termination II deglaciation. Using SEAMUS, histograms
of single specimen $\delta^{18}O$ values for discrete depths can also be explored, for example for sediment
core intervals with a median downcore age corresponding to the early Holocene (Fig. 2b), mid-
Holocene (Fig. 2c), Younger Dryas (Fig. 2d) and Late Glacial Maximum (Fig. 2e). This analysis
demonstrates the potential for the presence of single specimens with glacial climate values being
present in samples with an interglacial mean value. For example, in the early Holocene depth
interval (Fig. 2c), 15% of the simulated single specimens have a $\delta^{18}O$ value less than or equal to -
36‰. Of course, some sediment archives may have much higher lower SAR than the constant 10
cm/ka simulated in this example. The contribution of older specimens to a particular depth interval
is dependent upon a number of factors; temporal changes in SAR, BD, species abundance and the
susceptibility of older specimens to be broken/dissolved as a consequence of having been through
more bioturbation cycles (Rubin and Suess, 1955; Ericson et al., 1956; Emiliani and Milliman, 1966;
Barker et al., 2007). Using the SEAMUS model it is possible to run dynamic sediment scenarios to
investigate the influence of mixing of specimens of different ages upon interpretations based upon
single specimen analysis.





### 4.2 Analysing $^{14}$C calibration skill

As outlined earlier, it is possible to assign $^{14}$C activities to single specimens in the sedimentation

simulation based by using suitable records of the Earth's $\Delta^{14}$C history (e.g., *IntCal*). Subsequently,

SEAMUS uses the $^{14}$C activities of the specimens contained within each discrete depth to calculate

and expected laboratory $^{14}$C determination and measurement uncertainty. Using the *MatCal*

software, it is subsequently possible to calibrate the aforementioned $^{14}$C age, in combination with a

calibration curve and reservoir age estimate, to produce an expected calibrated age distribution. The

calibrated age distribution for the discrete depth can be compared with the true age distribution for

the discrete depth, as recorded by the simulation, to evaluate the skill with which current $^{14}$C dating

and calibration processes can reproduce the true age distribution of a particular sediment core slice.

A graphical representation of the aforementioned output for a discrete depth interval is shown in

Fig. 3, once again using the SEAMUS bioturbation simulation detailed in Section 3.1. This analysis

demonstrates that, for the applied simulation parameters and for the discrete depth interval

analysed in Fig. 3 (121-122 cm), the $^{14}$C calibration process would produce a median calibrated age

of 12.21 cal ka BP, whereas the true median age is 11.79 ka, meaning that there is a 420 year

difference between the two. Furthermore, the $^{14}$C calibration process produces a 95.45% credible

interval of 12.64 – 11.65 cal ka BP (a range of 990 cal yr), whereas the true 95.45% interval of the

single specimens within the simulation is 14.95-11.16 ka (a range of 3788 years), meaning that the

$^{14}$C dating and calibration process considerably underestimates (by some 2800 years) the age

uncertainty for this particular interval of simulated sediment core.  A Matlab script enabling users to

produce a figure similar to Fig. 3 is included within the tutorial script (*tutorial.m*) that is bundled with

SEAMUS. Users can subsequently explore downcore changes in the effectiveness of $^{14}$C dating to

accurately estimate true age under various dynamic simulation conditions, including: abundance

changes, SAR changes, bioturbation depth changes, reservoir age changes, as well as during periods

of dynamic $\Delta^{14}$C.

### 4.3 Investigating noise created by the picking process

When picking discrete-depth samples from discrete-depth specimen populations,

palaeoceanographers randomly pick whole specimens to produce a downcore mean signal. The

*seamus_pick* module can be used to test for random noise introduced upon the mean signal by the

picking process. The module can be repeatedly run with a set number of randomly picked whole

specimens per sample, and the resulting picking runs can be compared to an ideal picking run that

picks all available whole specimens for each discrete depth. Such an approach is investigated here,

once again using the same SEAMUS bioturbation simulation that was carried out in Section 3.1, for



picking scenarios each with one specimen per sample (Fig. 4a), two specimens per sample (Fig. 4b),
three specimens per sample (Fig. 4c), five specimens per sample (Fig. 4d), 10 specimens per sample
(Fig. 4e) and 20 specimens per sample (Fig. 4f). Such simulations can allow researchers to isolate and
quantify the effect of the picking process upon their downcore multi-specimen reconstructions for
their particular sediment core scenario. It can be noted that for the 10 cm/ka simulation carried out
here, that large sample sizes ($n \geq 10$) tend to produce downcore sampling runs close to the total
population mean (Figs. 4E and 4F), although the true spread of values is hidden. Furthermore, even
with larger samples sizes there is still the possibility for the generation of picking noise-induced
peak/trough values which could be erroneously interpreted as a precise indication of the timing of a
particular climate feature. In the case of very small sample sizes (Figs. 4A and 4B), researchers can
get an idea of the total spread of values within single core intervals. With advances in mass
spectrometry making the analysis of single specimens ever more routine and cost-effective, the ideal
approach in the future may involve exclusively analysing single specimens, with single specimen
values from discrete depths used to both estimate the signal distribution and calculate a downcore
mean signal, thus facilitating a 'best of both worlds' approach.
**4.4 Investigating noise created by absolute specimen abundance**
The interaction between total specimen abundance and bioturbation creates downcore noise in the
sedimentary record. In Fig. 5, the downcore, discrete-depth median age increase per centimetre for
three SEAMUS simulations all with an idealised constant SAR of 10 cm ka$^{-1}$ and constant BD of 10 cm
is shown, with the number of outputted specimens per centimetre being set differently for each
simulation, namely at 10$^{2}$ specimens per cm (Fig. 5a), 10$^{3}$ specimens per cm (Fig. 5b) and 10$^{4}$
specimens per cm (Fig. 5c). In all three scenarios the downcore, discrete-depth increase in median
age clusters around 100 years cm$^{-1}$, which is what would be expected in the case of 10 cm ka$^{-1}$
sediment core. As expected, the signal-to-noise ratio is higher in cases of higher abundance. An
interesting side-effect of a decreased signal-noise-ratio is the increased likelihood of the generation
of apparent age-depth reversals. For example, in the abundance scenario with 10$^{2}$ specimens cm$^{-1}$
(Fig. 5a), 21.7% of the discrete-depth (1 cm) age-depth points produce an apparent age-depth
reversal. Due to the fact that many age-depth modelling software packages often consider such age-
depth reversals as outliers (Blaauw and Christen, 2011; Lougheed and Obrochta, 2019),
palaeoceanographers should be aware that the apparent age-depth reversals generated by very
noisy downcore signals caused by low specimen abundance may result in age-depth models that are
biased towards young ages. Also, while palaeoceanographers often quantify relative abundance as a
ratio between different species, it is additionally important to quantify the absolute abundance of a




particular species being studied in the form of number of specimens per specific sediment volume,
as this can give clues regarding the expected signal to noise ratio ascertained from a discrete-depth
analysis.

**4.5 Investigating artefacts created by dynamic specimen abundance**

In the previous sections, scenarios involving constant specimen abundance were explored. SEAMUS
is specifically designed with the ability to process multiple temporally dynamic inputs. In Fig. 6, the
effect of temporally dynamic species abundance for a theorised "Species A" is studied, once again
using a scenario with a constant SAR of 10 cm/ka and constant BD of 10 cm. Past studies using
simpler mixing models have previously shown that the downcore $\delta^{18}O$ signal for particular species
can display offsets that are in fact an artefact of the interplay between abundance and bioturbation
(Löwemark and Grootes, 2004; Trauth, 2013). Here, the single-specimen SEAMUS simulation is used
to investigate the effects of abundance and bioturbation upon the age-depth signal produced by
single specimens. In this scenario SEAMUS is driven using a dynamic input with six temporal maxima
in Species A specimen flux centred upon 10, 16 18, 28, 32 and 36 ka ago (Fig. 6a). The resulting post-
simulation absolute abundance of Species A in the depth domain (Fig. 6b) a smoothed out / mixing
of the abundance peaks as a result of bioturbation. The interaction between dynamic abundance
and bioturbation also has consequences for the discrete-depth age-depth relationship of Species A.
For example, the downcore change in discrete-depth median age for Species A (Fig. 6c) is less noisy
(i.e. less likely to produce outliers) for intervals close to the absolute abundance peaks, but
negatively offset from the target discrete-depth median age change of 100 years per cm that would
be associated with the 10 cm/ka sediment core simulation. This would be manifested in an age-
depth reconstruction as an age-depth plateau near to an abundance peak.
Similarly, the 95.45% discrete-depth age range for Species A is much more constrained in the case of
depth intervals located close to the abundance peaks (Fig. 6d), but less representative of the median
age for the total sediment (all specimens), with Species A being biased towards too young ages (Fig.
6e). This bias is an interesting finding, seeing as it has long been assumed that pooled specimen
samples used for dating (e.g., [14]C dating) should be retrieved from abundance peaks (Keigwin and
Lehman, 1994; Waelbroeck et al., 2001; Galbraith et al., 2015). This assumption is largely based on
the fact that [14]C dates sampled from abundance peaks are younger than the immediately
surrounding sediment (Rafter et al., 2018). However, the SEAMUS simulation suggests that
abundance peaks can result in ages that are anomalously young when compared to the total
sediment (Fig. 6e).



**5.0 Conclusion**

Deep-sea sediment archives are subject to systematic bioturbation, which can complicate palaeoclimate reconstructions sourced from sediment cores. Complications can include artefacts and/or spurious offsets in $^{14}$C age other carrier signals (such as $\delta^{18}$O) sourced from multi-specimen samples. The SEAMUS model allows users to interactively investigate how such artefacts and/or spurious offsets can be attributed to the mixing of single specimens. The model is suitable for users who are investigating the downcore mean signal and how it is affected by dynamic changes in input variables. The model is especially interesting for researchers who are using single-specimen foraminifera analysis to quantify past changes in seasonality or multi-centennial amplitude in regional climate variability, as it can assist researchers in understanding the influence of bioturbation upon their results and the interpretation. The model is also useful as a teaching resource; for example, users can keep all but one input variable constant, and learn to understand the influence of dynamic changes in that particular input variable upon the downcore specimen record. Subsequently, multiple dynamic variables can be introduced.

**Code availability**

The SEAMUS model and accompanying interactive tutorial can be downloaded from the Zenodo public repository: https://doi.org/10.5281/zenodo.3251655

**Acknowledgements**

This work was funded by Swedish Research Council (*Vetenskapsrådet* – VR) Starting Grant number 2018-04992. Thanks to LOG for hosting me as a guest researcher in Wimereux. Brett Metcalfe is thanked for various discussions about the state-of-the-art of single specimen foraminifera analysis (see also the resources at www.brett-metcalfe.com).

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





Table 1. Approximate run times and Matlab memory use in the case of a 70 ka simulation run with
10 year iterations and core capacity of $10^2$, $10^3$ and $10^4$ specimens per cm. The runs carried out using
Matlab 2017b on a 64-bit system with 8GB of RAM and an Intel i7-2600 processor.

|  | $10^2$ specimens cm$^{-1}$ | $10^3$ specimens cm$^{-1}$ | $10^4$ specimens cm$^{-1}$ |
|---|---|---|---|
| *seamus_run* | 2.5 s / 0.62 GB | 19.7 s / 0.66 GB | 237.5 s / 1.15 GB |
| *seamus_pick* | 11.4 s / 0.61 GB | 13.2 s / 0.64 GB | 37.8 s / 0.99 GB |






**Figure captions (also included with figures)**

Figure 1. **(a)** NGRIP $\delta^{18}$O record (North Greenland Ice Core Project members, 2004) plotted using the
latest GICC05 timescale (Rasmussen et al., 2014; Seierstad et al., 2014), adjusted by 50 years so that
1950 BCE is equivalent to 'present'. **(b)** Result of SEAMUS run using the NGRIP $\delta^{18}$O data as temporal
input data. SEAMUS run settings are shown in the panel inset. Also shown is the average of ten runs
of TURBO2 (Trauth, 2013), based on the same NGRIP input data and using a SAR of 10 cm ka$^{-1}$ and a
constant BD of 10 cm.
Figure 2. **(a)** Log heat map (in greyscale) of downcore single specimen $\delta^{18}$O value probability in the
form of a 0.25‰ by 1 cm matrix, based on the single specimen data from the SEAMUS run displayed
in Fig 1B. The probability for each matrix element is calculated as the number of specimens for each
discrete depth within a given 0.25‰ range, divided by the total number of specimens contained
within the discrete depth. The natural logarithm of the probability is subsequently plotted, in order
to increase visibility of low probability areas in the heat map. Also shown (in orange) are the $\delta^{18}$O
values corresponding to the mean and 95.45% intervals for each discrete depth interval. **(b, c, d and
e)** Single specimen $\delta^{18}$O histograms for various discrete-depth intervals.
Figure 3. Example of using output from a SEAMUS simulation to estimate $^{14}$C calibration skill for a
particular discrete-depth subsample. The green histograms represent the SEAMUS simulation
output: on the x-axis the true age distribution of the discrete-depth single specimens (with the green
diamond corresponding to the median true age), and on the y-axis the $^{14}$C age distribution of the
single specimens (with the green diamond corresponding to the mean $^{14}$C age). All histograms are
shown using 100 ($^{14}$C) year bins. The orange probability distribution on the y-axis represents a
normal distribution corresponding to an idealised laboratory $^{14}$C analysis of the single specimens,
where the orange square corresponds to the expected mean laboratory $^{14}$C age. The orange
probability distribution on the x-axis represents the calibrated age distribution arising from the
calibration of the laboratory $^{14}$C analysis using *Marine13* (Reimer et al., 2013). Also shown, for
reference, are the *Marine13* calibration curve 1sigma (dark grey) and 2sigma (light grey) confidence
intervals. Simulation output shown in the figure is based on the SEAMUS run in Fig 1B, with $^{14}$C
activities assigned to single specimens according to *Marine13* with a constant ΔR of 0±0 $^{14}$C yr. For
the picking and calibration, all single specimens within the 121-122 cm discrete depth are picked,
and calibration is carried out using *MatCal* (Lougheed and Obrochta, 2016) with *Marine13* and a ΔR
of 0±0 $^{14}$C yr.





Figure 4. Estimating noise induced by subsample size during the picking process. Based on the
SEAMUS simulation in Fig. 1b, six sample size scenarios are considered: **(a)** one specimen per
sample; **(b)** two specimens per sample; **(c)** three specimens per sample; **(d)** five specimens per
sample; **(e)** ten specimens per sample; **(f)** 20 specimens per sample. In each scenario, the downcore
picking process is repeated 10 times, and each picking run is represented by a coloured line. Also
shown in all panels is the mean $\delta^{18}O$ value for all single specimens within discrete depth intervals
(black line) and 95.45% intervals (filled grey area).
Figure 5. Estimating downcore age-depth noise induced by absolute species abundance in three
scenarios all involving involving a constant SAR of 10 cm ka$^{-1}$ and constant bioturbation depth of 10
cm. In all three panels, the data points (circles) indicate the downcore discrete-depth median age
increase for each cm of core depth. Green circles correspond to positive downcore median age
change, while orange data points correspond to negative downcore median age change (i.e.
apparent age reversals). The horizontal black line in each panel denotes the perfect downcore age
change of +100 years cm$^{-1}$ that would be associated with a constant SAR of 10 cm ka$^{-1}$. The yellow
interval denotes the still-active BD (10 cm) at the core top. The signal-to-noise ratio (SNR) is also
computed for each scenario as the ratio between the summed squared magnitudes of the signal and
of the noise. The still-active BD at the core top is excluded from the SNR calculation. Three different
abundance scenarios are shown: **(a)** constant abundance of 10$^2$ specimens cm$^{-1}$. **(b)** constant
abundance of 10$^3$ specimens cm$^{-1}$. **(c)** constant abundance of 10$^4$ specimens cm$^{-1}$.
Figure 6. Investigating the effect of temporal changes in a species' abundance upon its discrete-
depth age-depth signal in the case of a simulated sediment core with a constant SAR of 10 cm ka and
constant BD of 10 cm. In all panels, the yellow interval denotes the still-active BD (10 cm) at the core
top. **(a)** The temporal abundance for a given species "Species A" used in the SEAMUS simulation,
inputted into the model as a fraction of the per timestep specimen flux. **(b)** The resulting simulated
downcore, discrete-depth (1 cm) absolute abundance (number of specimens) for Species A. Vertical
grey bands correspond to the depth of the abundance peaks. **(c)** The downcore, discrete-depth (1
cm) change in median age based on samples containing only Species A specimens. Green circles
denote downcore increase in discrete-depth apparent median age (i.e. positive apparent SAR) and
orange circles denote downcore decrease in discrete-depth median age (i.e. apparent age reversals).
The horizontal black line in each panel denotes the perfect downcore age change of +100 years cm$^{-1}$
that would be associated with a constant SAR of 10 cm ka$^{-1}$. **(d)** The 95.45% age range of for Species
A for each discrete 1 cm depth. **(e)** The offset between the median age of Species A (Med$_A$) and the





median age of all specimens (Med$_{all}$). Shown in the panel is Med$_A$-Med$_{all}$. The horizontal black line
represents corresponds to zero (i.e., no offset).



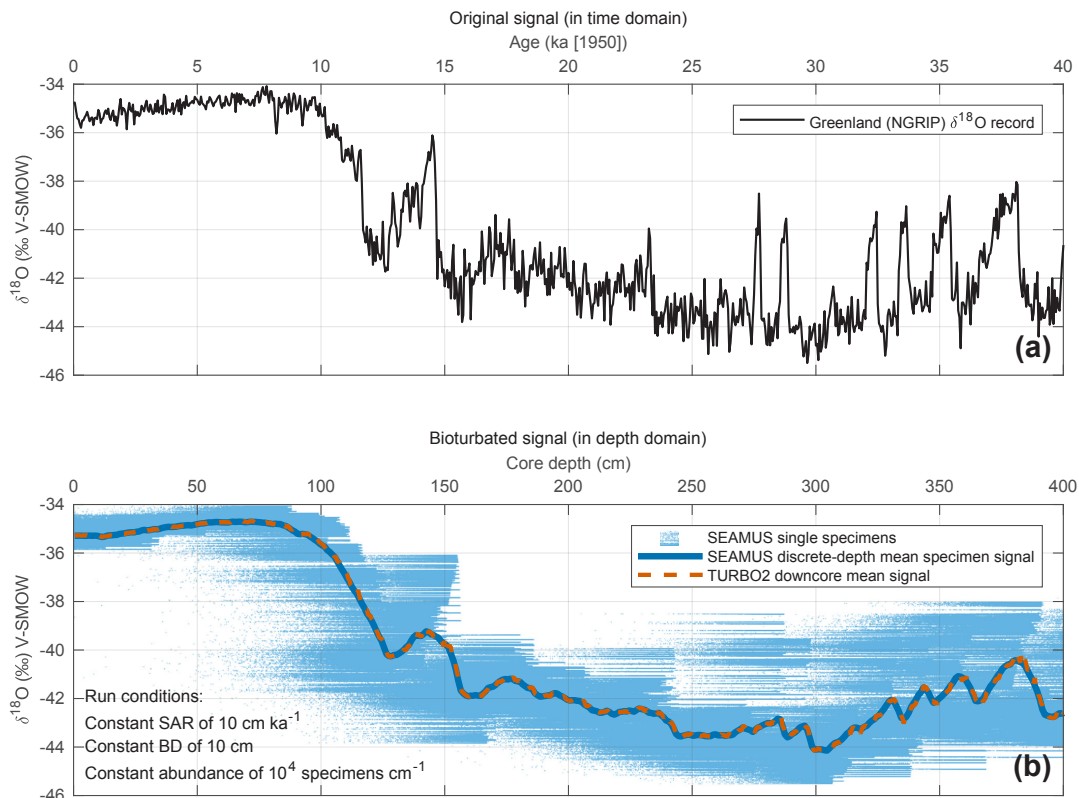

Figure 1. **(a)** NGRIP δ¹⁸O record (North Greenland Ice Core Project members, 2004) plotted using the latest GICC05 timescale (Rasmussen et al., 2014; Seierstad et al., 2014), adjusted by 50 years so that 1950 BCE is equivalent to 'present'. **(b)** Result of SEAMUS run using the NGRIP δ¹⁸O data as temporal input data. SEAMUS run settings are shown in the panel inset. Also shown is the average of ten runs of TURBO2 (Trauth, 2013), based on the same NGRIP input data and using a SAR of 10 cm ka⁻¹ and a constant BD of 10 cm.



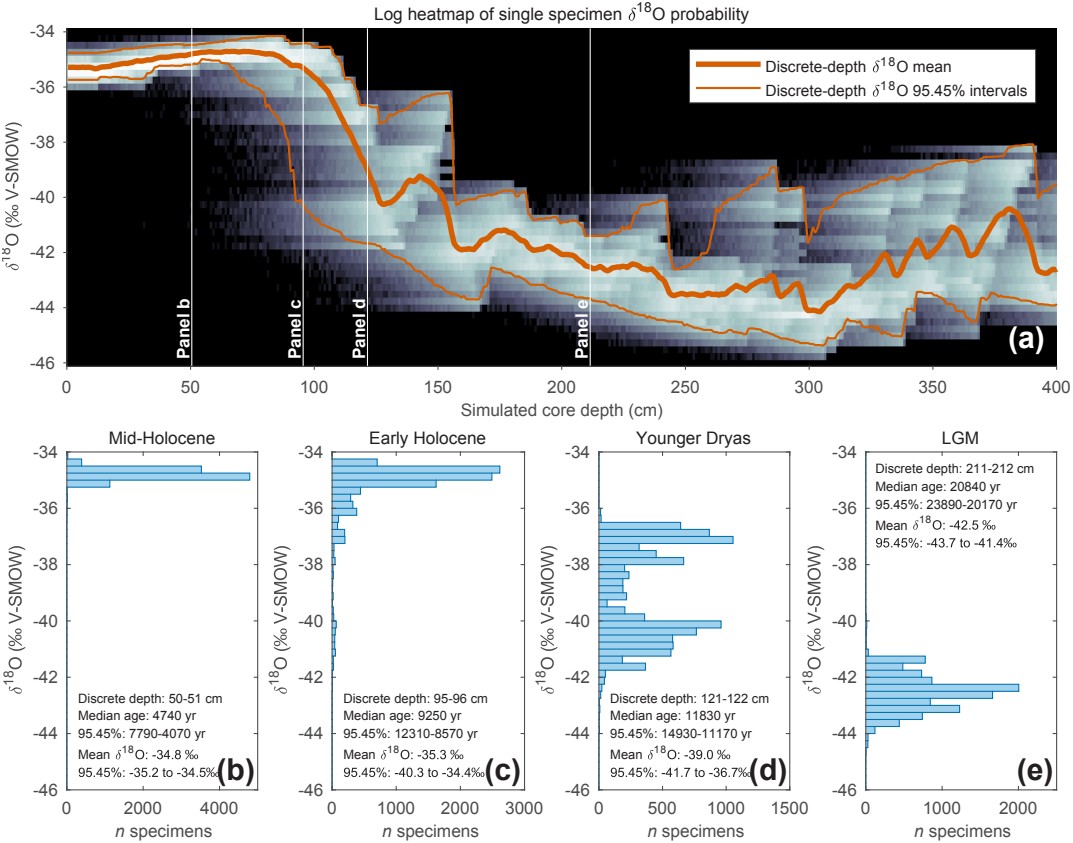

Figure 2. **(a)** Log heat map (in greyscale) of downcore single specimen δ[18]O value probability in the form of a 0.25‰ by 1 cm matrix, based on the single specimen data from the SEAMUS run displayed in Fig 1B. The probability for each matrix element is calculated as the number of specimens for each discrete depth within a given 0.25‰ range, divided by the total number of specimens contained within the discrete depth. The natural logarithm of the probability is subsequently plotted, in order to increase visibility of low probability areas in the heat map. Also shown (in orange) are the δ[18]O values corresponding to the mean and 95.45% intervals for each discrete depth interval. **(b, c, d and e)** Single specimen δ[18]O histograms for various discrete-depth intervals.

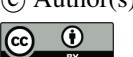



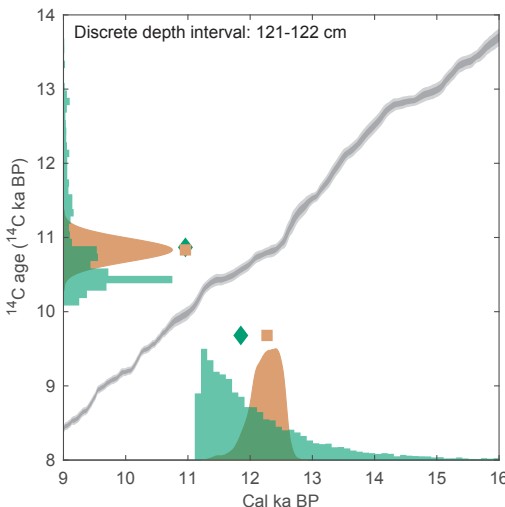

Figure 3. Example of using output from a SEAMUS simulation to estimate $^{14}$C calibration skill for a particular discrete-depth subsample. The green histograms represent the SEAMUS simulation output: on the x-axis the true age distribution of the discrete-depth single specimens (with the green diamond corresponding to the median true age), and on the y-axis the $^{14}$C age distribution of the single specimens (with the green diamond corresponding to the mean $^{14}$C age). All histograms are shown using 100 ($^{14}$C) year bins. The orange probability distribution on the y-axis represents a normal distribution corresponding to an idealised laboratory $^{14}$C analysis of the single specimens, where the orange square corresponds to the expected mean laboratory $^{14}$C age. The orange probability distribution on the x-axis represents the calibrated age distribution arising from the calibration of the laboratory $^{14}$C analysis using *Marine13* (Reimer et al., 2013). Also shown, for reference, are the *Marine13* calibration curve 1sigma (dark grey) and 2sigma (light grey) confidence intervals. Simulation output shown in the figure is based on the SEAMUS run in Fig 1B, with $^{14}$C activities assigned to single specimens according to *Marine13* with a constant ΔR of 0±0 $^{14}$C yr. For the picking and calibration, all single specimens within the 121-122 cm discrete depth are picked, and calibration is carried out using *MatCal* (Lougheed and Obrochta, 2016) with *Marine13* and a ΔR of 0±0 $^{14}$C yr.



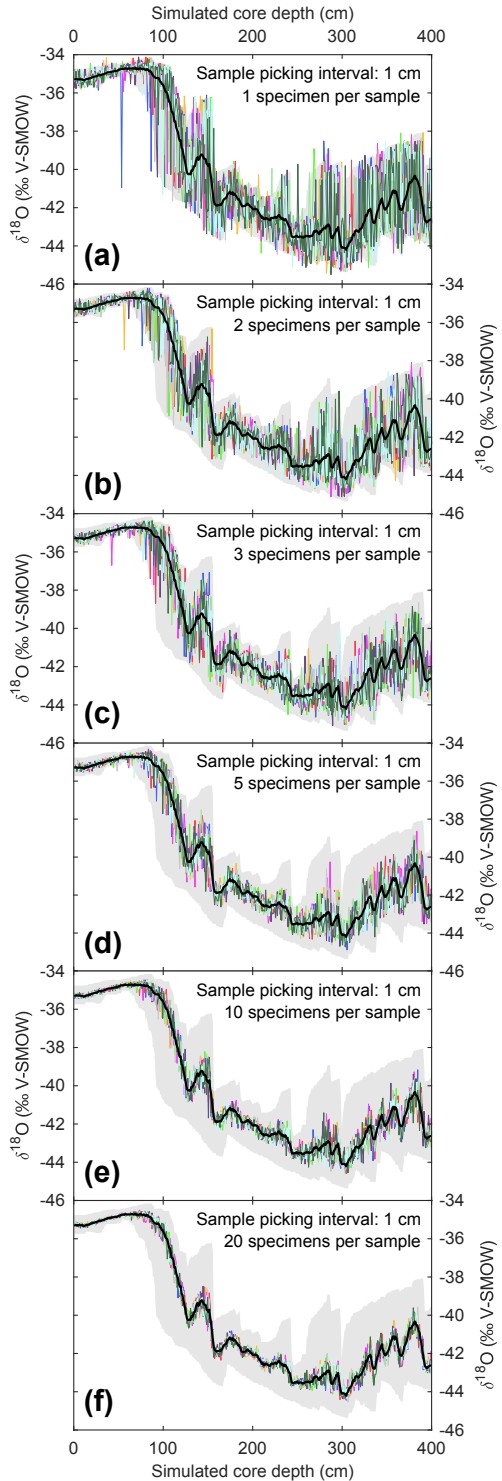

Figure 4. Estimating noise induced by subsample size during the picking process. Based on the SEAMUS simulation in Fig. 1b, six sample size scenarios are considered: **(a)** one specimen per sample; **(b)** two specimens per sample; **(c)** three specimens per sample; **(d)** five specimens per sample; **(e)** ten specimens per sample; **(f)** 20 specimens per sample. In each scenario, the downcore picking process is repeated 10 times, and each picking run is represented by a coloured line. Also shown in all panels is the mean $\delta^{18}O$ value for all single specimens within discrete depth intervals (black line) and 95.45% intervals (filled grey area).



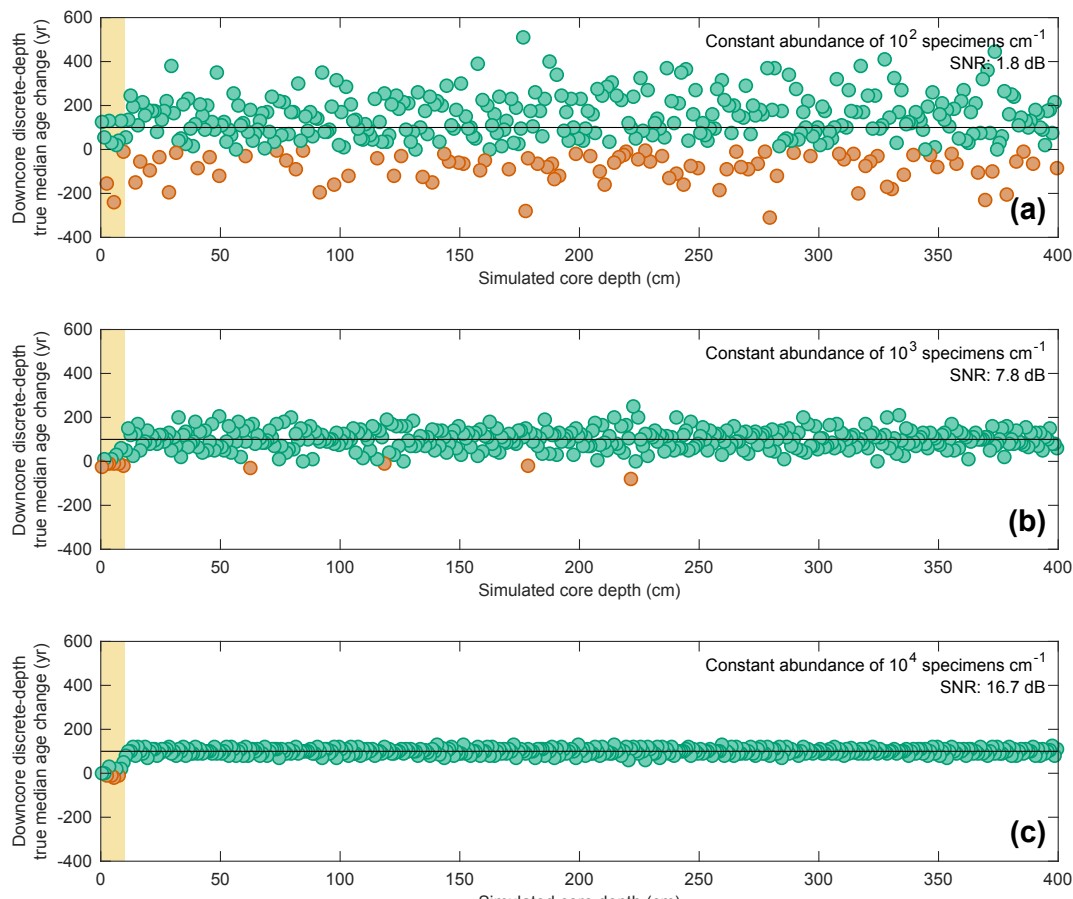

Figure 5. Estimating downcore age-depth noise induced by absolute species abundance in three scenarios all involving involving a constant SAR of 10 cm ka$^{-1}$ and constant bioturbation depth of 10 cm. In all three panels, the data points (circles) indicate the downcore discrete-depth median age increase for each cm of core depth. Green circles correspond to positive downcore median age change, while orange data points correspond to negative downcore median age change (i.e. apparent age reversals). The horizontal black line in each panel denotes the perfect downcore age change of +100 years cm$^{-1}$ that would be associated with a constant SAR of 10 cm ka$^{-1}$. The yellow interval denotes the still-active BD (10 cm) at the core top. The signal-to-noise ratio (SNR) is also computed for each scenario as the ratio between the summed squared magnitudes of the signal and of the noise. The still-active BD at the core top is excluded from the SNR calculation. Three different abundance scenarios are shown: **(a)** constant abundance of 10$^2$ specimens cm$^{-1}$. **(b)** constant abundance of 10$^3$ specimens cm$^{-1}$. **(c)** constant abundance of 10$^4$ specimens cm$^{-1}$.





Figure 6. Investigating the effect of temporal changes in a species' abundance upon its discrete-depth age-depth signal in the case of a simulated sediment core with a constant SAR of 10 cm ka and constant BD of 10 cm. In all panels, the yellow interval denotes the still-active BD (10 cm) at the core top. **(a)** The temporal abundance for a given species "Species A" used in the SEAMUS simulation, inputted into the model as a fraction of the per timestep specimen flux. **(b)** The resulting simulated downcore, discrete-depth (1 cm) absolute abundance (number of specimens) for Species A. Vertical grey bands correspond to the depth of the abundance peaks. **(c)** The downcore, discrete-depth (1 cm) change in median age based on samples containing only Species A specimens. Green circles denote downcore increase in discrete-depth apparent median age (i.e. positive apparent SAR) and orange circles denote downcore decrease in discrete-depth median age (i.e. apparent age reversals). The horizontal black line in each panel denotes the perfect downcore age change of +100 years cm$^{-1}$ that would be associated with a constant SAR of 10 cm ka$^{-1}$. **(d)** The 95.45% age range of for Species A for each discrete 1 cm depth. **(e)** The offset between the median age of Species A (Med$_A$) and the