# Peer review of "SEAMUS (v1.20): a $\Delta^{14}$ C-enabled, single-specimen sediment accumulation simulator"

_Geoscientific Model Development, 2019_

## Referee Comment (RC1) · Anonymous Referee #1 · 30 Aug 2019

**General comments.**

Lougheed et al present a stochastic simulation model that tracks the position of individual particles (e.g. Foraminifera) in an accumulating sediment as they are deposited and subsequently mixed by bioturbation (SEAMUS). A radiocarbon activity and geochemical proxy climate signal value can be attached to these particles so that the effects of mixing can be investigated on both age-depth models and proxy climate reconstructions from hypothetical sediment cores.

As SEAMUS explicitly tracks individuals, sediment accumulation rates, bioturbation depths, species flux rates and 14C reservoir ages can all be allowed to vary dynamically over time and the time integrated effects of these changes will be correctly modelled.

A similar analysis of age heterogeneity could be achieved using the existing sediment proxy forward modelling software "Sedproxy", by treating the 14C calibration curve as the input climate signal; however, sedproxy does not explicitly track individuals and relies on sampling from the theoretical age distribution of individuals in a given slice of sediment. To do this it has to assume that bioturbation depth and sedimentation rate are constant in the time period around the sample, although these can be different for different downcore samples. This approximation can lead to artefacts at the transition from periods of high to low sedimentation rate. Additionally, SEAMUS allows for the effects of changes in the absolute flux rates of Foraminifera to be investigated, whereas only relative changes in flux rate can be modelled using sedproxy. Furthermore, SEA-MUS keeps track of the number of bioturbation cycles each individual has experienced before it finally settles in the buried layers – this information is important for estimating the additional bias that can appear due to disintegration of the oldest specimens in a layer – and is not available from sedproxy.

Therefore, SEAMUS can be used to test/model scenarios that cannot by addressed without explicit modelling of the history of individuals. This comes at a cost of computing time, so it may not be the most appropriate tool in all situations, but it does represent a significant increase in realism and flexibility over existing software.

The effects of bioturbation on sediment proxy records, in particular the heterogeneity in age that adds noise to the signal and hidden uncertainty to the age model, are in general underappreciated, or if appreciated they are difficult to deal with. SEAMUS will provide a useful tool to check the possible size of these effects when interpreting data from a core, or planning a new sampling scheme.

**Specific comments.**

Although written for MATLAB, some of the SEAMUS code runs correctly on the open source GNU Octave language.

seamus_run required only minor alterations of the code to do with loading and saving

of files.

seamus_pick threw more errors but perhaps these are easy to fix for someone more familiar with Octave/MATLAB

If SEAMUS can be made compatible across MATLAB and Octave this would increase its availability.

**Technical comments.**

The manuscript was well written and I found only minor text errors:

Line 9 – "record" to "recorded"

Line 322 – "and expected" to "an expected"

---

## Author Comment (AC1) · 31 Aug 2019

I would like to thank reviewer 1 for taking the time to review and run the model and putting it into context with regards to other models. Since a second reviewer has yet to comment, I hereby reply to the first reviewer quickly in the interests of aiding the discussion forum.

The reviewer correctly describes that the main purpose, and unique feature, of the SEAMUS model is the tracking of single foraminifera during the history of a sediment core archive. This, as the reviewer correctly points out, does require more computational resources. I agree with the reviewer that the SEAMUS model has specific applications where it can be especially useful (e.g. single foraminifera analysis), whereas

other models (such as e.g. Sedproxy) might be especially useful in other applications (e.g. rapid and efficient computation of the bioturbated mean downcore signal). Additionally, it might be possible for users to use multiple types of models within a project: e.g. run Sedproxy and/or TURBO2 to rapidly experiment with mean downcore signal bioturbation for many different types of input scenarios, and then use SEAMUS to investigate single foraminifera relationships for specific chosen scenarios.

I would also like to thank the reviewer for taking the time to test run the SEAMUS model in Octave. I had assumed it would be too complex to run in Octave, but apparently it might be possible to get it to run. I will look into this and see if I can update the model so that it is compatible in both Matlab and Octave. In such a case I will update the table in the manuscript to also reflect computing times for Octave (which are usually much slower than Matlab). I agree that Matlab is not accessible for all users and that having an Octave version of SEAMUS would greatly increase availability of the model to the scientific community, especially in, e.g., developing countries. In future I hope to port the SEAMUS model to an open source language that is well optimised for rapid (vectorised) manipulation of very large matrices, such as the Julia language.

---

## Referee Comment (RC2) · Anonymous Referee #2 · 8 Oct 2019

This manuscript presents a a simple but useful model to simulate the impact of accumulation and bioturbation on sediment core signals.

The software appears to present a useful step forward for the community, and the manuscript explores a number of interesting implications/artefacts that idealised simulations with the model throw up. A focus is given to the interpretation of single foram geochemical analysis, which is timely. The provision of a script to analyse the implications of picking different numbers of forams from a sample is valuable.

By treating each added foram as a single element in an array, SEAMUS can follow a simple logical set of steps to simulate the bioturbation, and as such I have no concerns about the scripts themselves. The description of how the program operates, following on from the logical set of steps around which the script is built, is again clear and

logical.

I have no major concerns with this work, but rather a few minor suggestions, which may help make it valuable to a wider audience.

The first suggestion I would not expect to be acted upon here, but I think is useful to make in this forum. The community who undertake sediment-based paleoclimate analysis are perhaps increasingly, but still scantly familiar with programming. While Matlab is relatively accessible as programming languages go, there is a substantial cost associated with obtaining the software, which may prohibit an individual for making use of a tool like SEAMUS if they do not do other work in Matlab. While Octave provides an open source alternative to Matlab, and I support the other reviewer's suggestion of making the relatively minor changes required for the script to be able to run in Octave, use of a language like Python which is already available on most people's computers, and which could be built as a stand along program, would surely be very beneficial.

I suggest including a brief paragraph of the different sources of uncertainty in paleo-climate reconstructions (which I appreciate is a large topic), and identifying bioturbation within this. The reason for suggesting this is that the way that uncertainties are discussed in the latter part of the manuscript does not directly attribute these to bioturbation alone, and a user who treated this model as a black box (as is unfortunately sometimes the case), may miss this point and believe they are generating more fully assessing the uncertainty in their analysis.

Is there scope for turning the virtual picking simulator the other way round, i.e. use this approach to tell the user how many individuals they should be picking ahead of time? I appreciate that this can be achieved by playing around with the model, but it would be a simple addition to the code, which I would again anticipate would increase the audience for this work.

I wonder if it would be useful to bring some of the conclusions regarding the artefacts which can be generated by bioturbation into the abstract. While these are logical and

some discussed elsewhere, I found them thought provoking and very clear when presented in the context of the idealised model simulations, and I suspect that highlighting them may help convince people that they should be making use of a tool like SEAMUS.

Finally, two minor points about figure 2. Firstly, part A requires a color bar to be able to interpret it fully. Secondly, I found the caption to be considerably less readable than the rest of the manuscript. While it was possible to understand the figure from the main text and a bit of thinking, I found the figure caption actually confused rather than helped me.

---

## Author Comment (AC2) · 9 Oct 2019

It is not always easy to find willing referee for these types of technical papers, so I would like to thank the referee for taking the time to review the manuscript and scripts. I would also like to thank the referee for consulting the review of the other referee, thereby taking advantage of the discussion aspect of GMD.

I agree with the referee that not everyone has access to the Matlab environment (especially those in developing countries) and I will strive to make the simulation Octave compatible if possible. While Octave is not pre-installed on most computers, it is free to download. (Similarly, anybody with python pre-installed would need to additionally download e.g. numpy). In future the model could be ported to another language and/or

a GUI front end could be constructed, seeing as programming is not widespread within the palaeo community, as the referee notes. The model is of course published open source so I'd be open to a python specialist helping me create a python fork. Personally I am keeping an eye on the Julia language at the moment, but it doesn't seem to be fully mature yet.

Referee comment: "Is there scope for turning the virtual picking simulator the other way round, i.e. use this approach to tell the user how many individuals they should be picking ahead of time? I appreciate that this can be achieved by playing around with the model, but it would be a simple addition to the code, which I would again anticipate would increase the audience for this work."

By number of individuals I assume the referee means the minimum number of individuals that should be picked to result in a discrete-depth mean reproducibility that would be equal or better than the machine measurement uncertainty. This would of course depend upon the particular proxy method being used. I will look into this. I'd reiterate, though, that one of the main motivations of creating the SEAMUS model was to highlight the fact that the spread of values contained within a particular core depth can be much greater than the machine error. As is written (near line 361): "With advances in mass spectrometry making the analysis of single specimens ever more routine and cost-effective, the ideal approach in the future may involve exclusively analysing single specimens, with single specimen values from discrete depths used to both estimate the signal distribution and calculate a downcore mean signal, thus facilitating a 'best of both worlds' approach."

A small discussion about the possible generation of downcore artefacts can also be included. Thank you for this suggestion. It may be in there already but should be brought more to the fore.

I will improve Figure 2 (add a colour bar and make a better caption).

Thanks again for your helpful review.

a+ Bryan Lougheed

---

## Author Response (AR1)

Uppsala, Sweden.
November 22, 2019.

Dear Paul Halloran,

Thank you for considering my manuscript. Please find attached the revised version of the manuscript "SEAMUS: a $\Delta^{14}$C-enabled, single-specimen sediment accumulation simulator". The referees have raised some valuable suggestions which have improved both the software and the manuscript. Below, I sum up the main suggestions and the action that has been taken:

Referee #1 main suggestion: Make the software Octave compatible (Referee #2 concurred)
Action: The software has now been made fully Octave compatible and the manuscript has been updated to reflect this fact. A separate tutorial .m f[ile optimised for Octave users has also been created. This upgrade involved some work, but was well worth it to make the software more accessible to the wider community.

Referee #2 first main suggestion: More information about various sources of error, to make the reader aware that bioturbation is not the only source.
Revision action: In the abstract, introduction and conclusion I have now emphasised that the SEAMUS bioturbation model can be combined with other resources (such as proxy and ecological models) to attain a complete picture of the total uncertainty involved in palaeoclimate reconstructions retrieved from sediment archives.

Referee #2 second main suggestion: Add colour bar to Figure 2 and improve caption.
Revision action: These improvements have now been undertaken. Figure 2 now also uses a regular heatmap instead of a logarithmic one, to optimise legibility of the colour bar for *H. sapiens*.

Referee #2 third main suggestion: *Is there scope for turning the virtual picking simulator the other way round, i.e. use this approach to tell the user how many individuals they should be picking ahead of time? I appreciate that this can be achieved by playing around with the model, but it would be a simple addition to the code, which I would again anticipate would increase the audience for this work.*
Revision action: I have put quite some thought into how this and came to the conclusion that "minimum number of individuals" to be picked, while indeed useful, is dependent on the type of proxy being studied, the level of noise deemed acceptable by the end-user, the desired depth resolution, etc. As such, I decided against adding such a feature as it might result in the end-user relying on the model as a black box recommendation of number of foraminifera to pick, as the referee will appreciate. So I would encourage the end-user to iteratively explore different picking scenarios, as outlined in Section 4.3.

Other revision actions:
I removed the "(v1.0)" from the title, as the software has since moved on in version, and the manuscript should reflect the software in general, rather than a specific version.

I also decided to remove the supplemental tables with descriptions of the input and output variables. Users can simply refer to the function documentation included in the script files. The script documentation will always be up to date.

Thanks again for considering my manuscript. Please don't hesitate to contact me should you require further information.

Kind regards,
Bryan Lougheed

[revised manuscript text omitted]
250     error of $\pm$30 $^{14}$C yr, and a $\underline{\text{determination with the } F^{14}C \text{ value } e^{(blankvalue-1)/-8033}}$ $\underline{\text{(i.e. one }^{14}C\text{ yr younger than}}$  the blank value$\underline{)\text{ is assigned}}$  an error of $\pm$200 14C yr $\underline{\text{(this}}$ $\underline{\text{value can be customised by the user in the input parameters). Errors (in }^{14}C\text{ yr) for intermediate dates}}$ $\underline{-}$  are linearly $\underline{\text{interpolated to } F^{14}C\text{. The } MatCal}$   (Lougheed and Obrochta, 2016) calibration software $\underline{\text{
[revised manuscript text omitted]